

# Inversion model of soil salinity in alfalfa covered farmland based on sensitive variable selection and machine learning algorithms

Hong Ma[1,2,3,*], Wenju Zhao[1,3,*], Weicheng Duan[1,3], Fangfang Ma[1,3], Congcong Li[1,3] and Zongli Li[4]

[1] College of Energy and Power Engineering, Lanzhou University of Technology, Lanzhou, China
[2] JiuQuan Vocational Technical College, JiuQuan, China
[3] Ministry of Agriculture and Rural Affairs Smart Agriculture Irrigation Equipment Key Laboratory, Lanzhou, China
[4] General Institute for Water Resources and Hydropower Planning and Design, Ministry of Water Resources, Beijing, China
* These authors contributed equally to this work.

Corresponding author
Wenju Zhao, wenjuzhao@lut.edu.cn

## ABSTRACT

**Purpose:** Timely and accurate monitoring of soil salinity content (SSC) is essential for precise irrigation management of large-scale farmland. Uncrewed aerial vehicle (UAV) low-altitude remote sensing with high spatial and temporal resolution provides a scientific and effective technical means for SSC monitoring. Many existing soil salinity inversion models have only been tested by a single variable selection method or machine learning algorithm, and the influence of variable selection method combined with machine learning algorithm on the accuracy of soil salinity inversion remain further studied.

**Methods:** Firstly, based on UAV multispectral remote sensing data, by extracting the spectral reflectance of each sampling point to construct 30 spectral indexes, and using the pearson correlation coefficient (PCC), gray relational analysis (GRA), variable projection importance (VIP), and support vector machine-recursive feature elimination (SVM-RFE) to screen spectral index and realize the selection of sensitive variables. Subsequently, screened and unscreened variables as model input independent variables, constructed 20 soil salinity inversion models based on the support vector machine regression (SVM), back propagation neural network (BPNN), extreme learning machine (ELM), and random forest (RF) machine learning algorithms, the aim is to explore the feasibility of different variable selection methods combined with machine learning algorithms in SSC inversion of crop-covered farmland. To evaluate the performance of the soil salinity inversion model, the determination coefficient ($R^2$), root mean square error (RMSE) and performance deviation ratio (RPD) were used to evaluate the model performance, and determined the best variable selection method and soil salinity inversion model by taking alfalfa covered farmland in arid oasis irrigation areas of China as the research object.

**Results:** The variable selection combined with machine learning algorithm can significantly improve the accuracy of remote sensing inversion of soil salinity. The performance of the models has been improved markedly using the four variable

selection methods, and the applicability varied among the four methods, the GRA variable selection method is suitable for SVM, BPNN, and ELM modeling, while the PCC method is suitable for RF modeling. The GRA-SVM is the best soil salinity inversion model in alfalfa cover farmland, with $R_v^2$ of 0.8888, $RMSE_v$ of 0.1780, and RPD of 1.8115 based on the model verification dataset, and the spatial distribution map of soil salinity can truly reflect the degree of soil salinization in the study area.
**Conclusion:** Based on our findings, the variable selection combined with machine learning algorithm is an effective method to improve the accuracy of soil salinity remote sensing inversion, which provides a new approach for timely and accurate acquisition of crops covered farmland soil salinity information.

# INTRODUCTION

Soil salinization has become one of the most serious soil problems in arid and semi-arid areas, leading to soil consolidation, fertility decline, and crop yield reduction, thus restricting the regional protection of the ecological environment and sustainable economic development (*Dahlawi et al., 2018*; *Van Zelm, Zhang & Testerink, 2020*; *Hassani, Azapagic & Shokri, 2021*). According to statistics, the existing saline soil area in China is about 3,600.0 million $hm^2$, which is prominent in the northwest arid area where the saline soil accounts for 69.03% of the area, limiting healthy agriculture development in the arid area (*Yang et al., 2022*; *Mishra et al., 2023*; *Pan et al., 2023*). Therefore, rapidly and accurately obtaining information on soil salt content is a prerequisite for effectively evaluating and managing soil salinization and reasonably utilizing saline soil (*Masoud et al., 2019*; *Devkota et al., 2022*). Though traditional methods of obtaining information on soil salt content, such as field sampling, are relatively accurate, they are difficult to realize large area dynamic monitoring of salinization area due to its long cycle, high cost, complex process and poor real-time performance (*Khasanov et al., 2022*; *Abdalla et al., 2022*). Remote sensing technology has become an important way to monitor soil salinity content in large scale because of its large area synchronous observation, high timeliness and powerful data integration (*Wu et al., 2021*; *Mohamed et al., 2023*).

As an important means of remote sensing, uncrewed aerial vehicle (UAV) remote sensing has the advantages of real-time image transmission, low operating cost and flexibility (*Maes & Steppe, 2019*). The technique is a strong supplement to satellite remote sensing, which has been widely used recently in the dynamic monitoring of soil salinization (*Zhao et al., 2022b*; *Zhao et al., 2023*; *Salgado et al., 2023*). *Zhang et al. (2022)* established the soil salinity inversion model by calculating spectral index from UAV multispectral remote sensing images, which enabled dynamic monitoring of soil salinity at different depths under different coverage of the study area. Moreover, *Zhu et al. (2022)* effectively predicted and characterized the spatial distribution pattern of soil salinity using UAV visible and near-infrared spectra, which also measured salinity data combined with

the commonly used modeling methods. *Ivushkin et al. (2019)* studied the potential of UAV remote sensing in monitoring plant salt stress using three different UAV sensors combined with vegetation index and canopy temperature parameters. These studies show that the soil salinity inversion model established using UAV remote sensing data can quickly and effectively monitor the degree of soil salinization; however, its accuracy is quite low ($R^2$ of 0.65).

Therefore, scholars have further explored various methods of improving the accuracy of remote sensing inversion, including sensitive variable selection, which is important for the accuracy of remote sensing inversion. The common variable selection methods include the filter methods, wrapper methods, and embedded methods (*Ma et al., 2020*; *Cui et al., 2022*). *Lin et al. (2016)* used the PCC to effectively prevent the loss of correlation information between remote sensing data and soil organic matter, thus improving the inversion accuracy of soil organic matter. Furthermore, *Jia et al. (2021)* used the GRA to screen the sensitive variables for the soil PH inversion model and achieved better inversion accuracy. *Cevoli et al. (2022)* used the VIP to screen the spectral bands and found that reducing the number of variables would not affect the accuracy of the model, indicating that the VIP variable selection method could effectively remove redundant information. Variable selection can optimize the inverse model and improve the accuracy of the inversion models. However, the current variable selection methods are mainly used to measure parameters such as soil organic matter (*Cao et al., 2020*), soil pH (*Miao, Li & Lu, 2018*), and vegetation water content (*Han et al., 2019*) but have not been used for soil salinity analysis.

In the research method of using UAV remote sensing data to invert soil salinity information, the variables sensitive to soil salinity are generally selected by means of spectral transformation, spectral index improvement or spectral index selection, so as to improve the performance of the model (*Wang et al., 2018a*; *Zhang et al., 2019a*). Therefore, the sensitive variable selection is very important for the construction of soil salinity inversion model. *Zhao et al. (2022b)* used the PCC to select the spectral index with high correlation to construct the inversion model of soil salinity in the surface layer of different vegetation covers, and the model inversion results can better reflect the real soil salinity content in the region; *Chen et al. (2020)* constructed the soil salinity inversion model of field sunflower at different growth stages and different soil depths based on different machine learning algorithms on the basis of using GRA to screen the spectral index. *Jia et al. (2024)* constructed vegetation and salinity index using remote sensing data, and constructed a soil salinity monitoring model by selecting the vegetation and salinity index using the VIP algorithm combined with the extreme learning machine; *Zhao et al. (2022a)* used the SVM-RFE algorithm to screen the spectral index, and constructed soil salinity inversion models with different crop coverage and different depths. Although variable selection methods combined with machine learning algorithms have been applied by scholars to the construction of soil salinity inversion models and have achieved good results, there are significant differences between different variable selection methods in improving model performance, increasing computational efficiency, and reducing the impact of redundant information. Therefore, based on the UAV remote sensing platform,

the comparative study of different variable selection methods combined with different machine learning algorithms for soil salinity inversion in vegetation-covered farmland needs to be further deepened, trying to explore: which variable selection method is the most effective? Which variable selection methods combined with machine learning algorithms can effectively improve the prediction accuracy of the model?

Considering the foregoing, this study used PCC, GRA, VIP, and SVM-RFE to screen the variables of the constructed 30 spectral indexes, constructed 20 soil salinity inversion models based on support vector machine regression (SVM), back propagation neural network (BPNN), extreme learning machine (ELM), and random forest (RF). This study aimed to: (1) determine the spectral index suitable for soil salinity inversion in alfalfa covered farmland in arid oasis areas; (2) compare and evaluate the effectiveness of PCC, GRA, VIP, and SVM-RFE variable selection methods in selecting sensitive variables, whether they can improve the accuracy of soil salinity inversion; (3) evaluate the predictive accuracy of different variable selection methods coupled with machine learning algorithms, and determine the best variable selection method and soil salinity inversion model. To provide a new approach for timely and accurate acquisition of crops covered farmland soil salinity information.

## MATERIALS AND METHODS

### Study area

In this article, the arid oasis irrigation area in China is selected as the study area (Fig. 1). This region is located at $98°20'\sim99°18'$E, $39°10'\sim40°15'$N, with temperate continental arid climate, dry and little rain, strong evaporation, average precipitation of 72.6 mm and average evaporation of 2,184.0 mm. These characteristics make it a typical "no irrigation, no agriculture" area, which main crops are grain economic crops such as alfalfa, wheat and corn. According to the survey, soil salinization in the region is spreading from spotty to patchy, accompanied by problems such as water shortage and fragile ecological environment. This article conducts research on inversion model of soil salinity in alfalfa covered farmland based on sensitive variable selection and machine learning algorithms by taking alfalfa covered farmland as the research object. The study and the corresponding results have crucial practical significance and research value for developing precision agriculture.

### Research methods

#### Data source and preprocessing

(1) UAV multispectral data acquisition and preprocessing

The UAV remote sensing platform used in this study is the DJI P4 multispectral version with a visible light camera and five multispectral cameras (blue 450 nm, green 560 nm, red 650 nm, near-infrared 840, and nmred edge 730 nm). The work was conducted at the Bianwan Farm in the Taolai River Basin on June 15, 2023, and the UAV multispectral image data was acquired between 11:00 am and 14:00. During the experiment, the weather was clear and cloudless, the flight altitude was 75.0 m, the average speed was 4.0 m/s, the
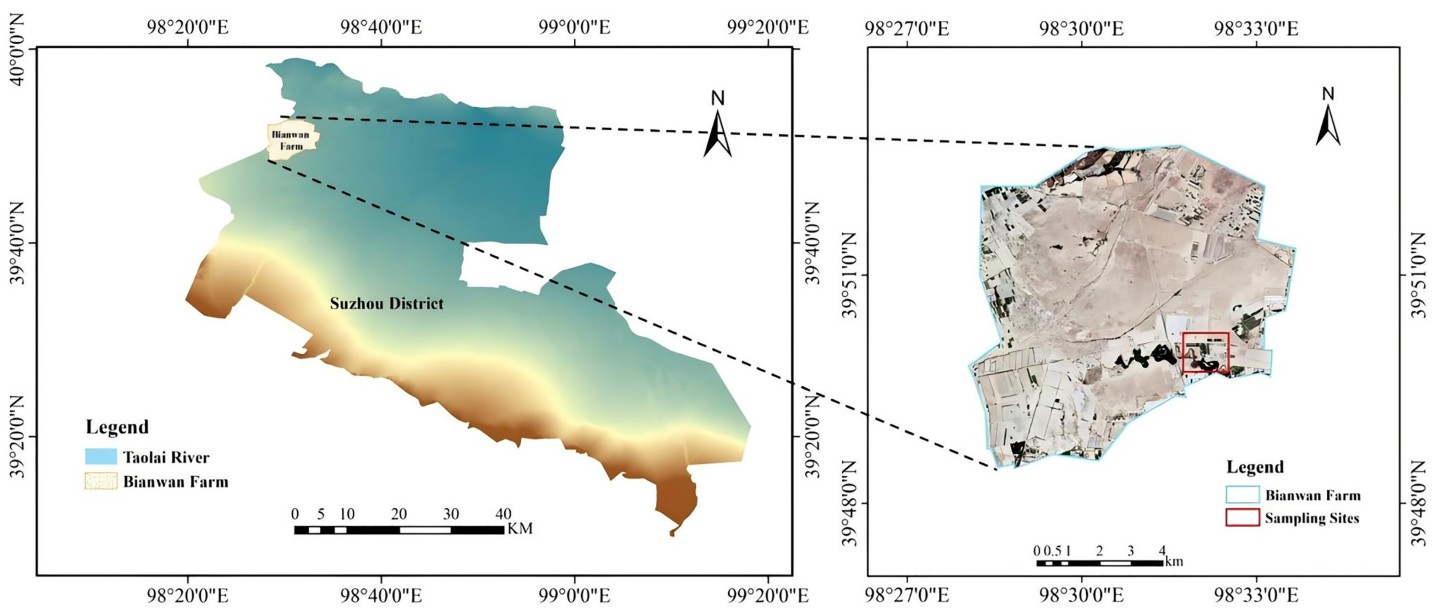

**Figure 1** Study area and diagram.

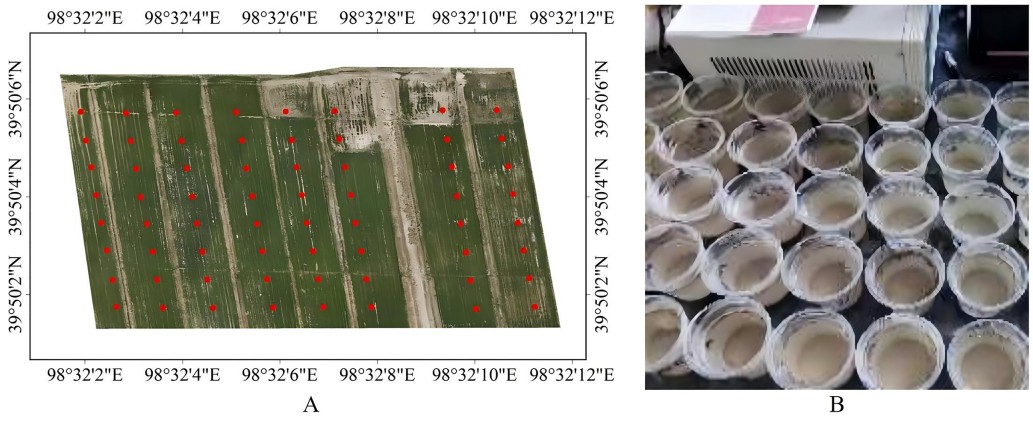

A          B

**Figure 2** SSC data acquisition. (A) Sampling point distribution diagram. (B) Soil salinity measurement.

ground resolution was 4.0 cm, and the heading and sidetrack overlap was set to 75%, and standard whiteboards were used to calibrate the data. Finally, the acquired multispectral images were imported into DJI Wisdom for image correction, cropping, and other preprocessing procedures, which used to explore the feasibility of different variable selection methods combined with machine learning algorithms in improving the accuracy of SSC inversion in crop-covered farmland.

(2) Soil salinity content data acquisition

To ensure temporal consistency between the UAV multispectral remote sensing image and soil salinity data, the experiment sampled field soil samples on June 15, 2023 at the Bianwan Farm in the Taolai River Basin, and the data collection time was the flowering

**Table 1 The spectral index and calculation formula.**

| Vegetation index | Calculation formula | Reference | Salinity index | Calculation formula | Reference |
|---|---|---|---|---|---|
| Normalized difference vegetation index (NDVI) | $(NIR - R)/(NIR - R)$ | *Zhao et al. (2022b)* | Normalized difference soil index (NDSI) | $(R - NIR)/(R + NIR)$ | *Miao, Li & Lu (2018)* |
| Differential vegetation index (DVI) | $NIR - R$ | | Brightness index (BI) | $(NIR^2 + R^2)^{0.5}$ | *Bouaziz et al. (2018)* |
| Enhanced vegetation index (EVI) | $2.5 (NIR - R)/(NIR + 6R - 7.5B + 1)$ | | Salinity index (S1) | $B/R$ | |
| Ratio vegetation index (RVI) | $NIR/R$ | | Salinity index (S2) | $(B - R)/(B + R)$ | |
| Triangular vegetation index (TVI) | $0.5 [120 (NIR - G) - 200 (R - G)]$ | | Salinity index (S3) | $(G * R)/B$ | |
| Normalized difference greenness vegetation index (NDGI) | $(G - R)/(G - R)$ | *Mwinuka et al. (2022)* | Salinity index (S4) | $(B * R)^{0.5}$ | |
| Greenness normalized difference vegetation index (GNDVI) | $(NIR - G)/(NIR + G)$ | *Li et al. (2022)* | Salinity index (S5) | $(B * R)/G$ | |
| Phenological normalized difference vegetation index (PNDVI) | $[NIR - (G + R + B)]/[NIR + (G + R + B)]$ | | Salinity index (S6) | $(NIR * R)/G$ | |
| Greenness vegetation index (GVI) | $(NIR/G)/(R/G)$ | *Sun, Song & Lu, 2022* | Salinity index1 (SI1) | $(G * R)^{0.5}$ | |
| Soil adjusted vegetation index (SAVI) | $1.5 (NIR - R)/(NIR + R + 0.5)$ | *Zhang et al. (2022)* | Salinity index2 (SI2) | $(NIR^2 + G^2 + R^2)^{0.5}$ | |
| Improved soil adjusted vegetation index (MSAVI) | $0.5 [2NIR + 1 - ((2NIR + 1)^2 - 8 (NIR - R))^{0.5}]$ | | Salinity index3 (SI3) | $(G^2 + R^2)^{0.5}$ | |
| Canopy salinity responsive vegetation index (CRSI) | $[((NIR * R) - (G * B))/((NIR * R) + (G * B))]^{0.5}$ | | Salinity index4 (SI4) | $[(B - G)^2 + (G - R)^2]^{0.5}$ | |
| Atmospheric impedance vegetation index (ARVI) | $(NIR - 2R + B)/(NIR + 2R - B)$ | | Salinity index (SI) | $(B * R)^{0.5}$ | *Khan et al. (2005)* |
| Renormalized difference vegetation index (RDVI) | $[(NIR - R)/(NIR + R)]^{0.5}$ | *Zhang et al. (2019c)* | Salinity index (SI-T) | $100 (R-NIR)$ | *Weerakoon (2017)* |
| Optimized soil adjusted vegetation index (OSAVI) | $1.16 (NIR - R)/(NIR + R + 0.16)$ | | Comprehensive spectral response index (COSRI) | $(B + G)/(NIR + R) * NDVI$ | *Kumar, Gautam & Saha (2015)* |

**Note:**
B, G, R, NIR, and Rededge denote the spectral reflectance values at the wavelengths of 450, 560, 650, 840, and 730 nm, respectively.

period of alfalfa growth stage, and the sampling areas of alfalfa-covered land was 54,000 m$^2$ (Fig. 2). Sixty-four sampling points were evenly distributed in the sampling area, and soil samples from 0–15 cm were collected using soil augers, and information on the location of each point was recorded, the longitude and latitude of the sampling points are available in the Files S5. The location information of each point was recorded. Each soil sample with weight equal to 30 g was collected into an aluminum box, oven-dried for 8 h, cooled and then ground and sieved (pore size 2 mm). Distilled water of volume equal to 150 ml was added to the sieved soil samples and stirred thoroughly, and after a few hours of standing, the conductivity of the soil solution was determined using a Raymag DJS-1C conductivity meter. The soil salt content (SSC, %) was calculated according to the empirical formula SSC = 0.2882EC$_{1:5}$ + 0.0183 (*Zhao et al., 2022a*).

### Spectral index construction

Crops on saline soil in arid areas usually exhibit sparse coverage, so using only salinity index and vegetation index could not effectively monitor the degree of soil salinization. Therefore, we selected 30 spectral indexes (15 for vegetation and another 15 for salinity) as the input variables of the salinity inversion model (Table 1).

### Sensitive variable selection method

(1) Pearson correlation coefficient

Pearson correlation coefficient (PCC) was proposed by Karl Pearson in the late 19th century as a measure of the linear relationship between two variables (*Pearson, 1895*). The following is the calculation formula:

$$r = \frac{\sum (x_i - \bar{x})(y_i - \bar{y})}{\sqrt{\sum (x_i - \bar{x})^2 (y_i - \bar{y})^2}} \tag{1}$$

where $x_i$ and $y_i$ are the observed values of two variables X and Y, $\bar{x}$ and $\bar{y}$ are the mean values of variables X and Y.

(2) Gray relational analysis

Gray correlation analysis (GRA) method was proposed by Professor Deng Julong in the 1980s. As an analysis method based on grey system theory, which used to evaluate the relationship strength between system variables (*Deng, 1989*). The following is the calculation formula:

$$GCD = \frac{1}{m} \sum_{t=1}^{m} \beta[x_0(t), x_i(t)] \tag{2}$$

where $\beta(x_0(t), x_i(t)) = \frac{\Delta(min) + \rho\Delta(max)}{\Delta_{0i}(k) + \rho\Delta(max)}$; $\rho$ is the determining coefficient, 0.5.

(3) Variable projection importance

Variable importance projection (VIP) algorithm was proposed by *Wold, Johansson & Cocchi (1993)*. As a variable selection method based on partial least squares algorithm,

which used to evaluate the importance of each variable in the model (*Wold, Johansson & Cocchi, 1993*). The following is the calculation formula:

$$VIP = \sqrt{m \times \frac{\sum_{i=1}^{n} R_d(Y,t)W^2}{\sum_{i=1}^{n} R_d(Y,T)}} \tag{3}$$

where m for the number of independent variables, n is the number of components; t for the selected independent variables, $R_d(Y,t)$ for the interpretation degree of the dependent variable components, and $W^2$ designates the effect of variables in each component. The VIP value greater than one shows a strong relationship between the independent and dependent variables.

(4) Support vector machine recursive feature elimination

Support vector machine recursive feature elimination (SVM-RFE) algorithm was proposed by *Guyon et al. (2002)*. As a backward iterative recursive sensitive variable selection method based on support vector machine, it mainly eliminates the least important features by recursively training the support vector machine model, so as to obtain the optimal feature subset (*Guyon et al., 2002*).

### Construction and accuracy evaluation of the salinity inversion model

The collected 60 samples were divided in a ratio of 2:1, with 40 samples as the modeling set and the remaining 20 as the verification set. Taking the spectral index as the model input independent variable and their corresponding soil salt content as the dependent variable, on the basis of variable selection, we constructed 20 soil salinity inversion models based on SVM, BPNN, ELM and RF. Model accuracy was evaluated using the determination coefficient ($R^2$), root means square error (RMSE), and performance deviation ratio (RPD) (*Chen et al., 2022*). The following is the calculation formula:

$$R^2 = \frac{\sum_{i=1}^{n} (\hat{y}_i - \bar{y}_i)^2}{\sum_{i=1}^{n} (y_i - \bar{y}_i)^2} \tag{4}$$

where the closer $R^2$ is to 1, the higher the accuracy of the model fit.

$$RMSE = \sqrt{\frac{\sum_{i=1}^{n} (\hat{y}_i - y_i)^2}{n}} \tag{5}$$

where the closer RMSE is to 0, the higher the prediction accuracy of the model.

$$RPD = \frac{SD}{RMSE} \tag{6}$$

where RPD > 2.0 indicates good model capability, $1.4 \leq RPD \leq 2.0$ indicates a rough quantitative prediction ability of the model, and RPD < 1.4 shows that the model's predictive power is unreliable.

**Table 2 The statistical analysis of SSC.**

| Sample | | Number of samples | | | Salt content | | | | | |
|---|---|---|---|---|---|---|---|---|---|---|
| | | Mild salinization soil | Moderate salinization soil | Severe salinization soil | Maximum (%) | Minimum (%) | Average (%) | Standard deviation | Variance | Coefficient of variation |
| Alfalfa covered land | Total sample 60 | 15 | 39 | 6 | 0.76 | 0.37 | 0.47 | 0.322 | 0.104 | 0.20 |
| | Modeling set 40 | 7 | 28 | 5 | 0.76 | 0.37 | 0.49 | 0.339 | 0.115 | 0.21 |
| | Verification set 20 | 8 | 11 | 1 | 0.62 | 0.38 | 0.44 | 0.252 | 0.064 | 0.17 |

# RESULTS

## Statistical analysis of SSC

The 60 soil samples used for salt content analysis were divided into three groups, namely, mild salinization soil (<0.4%), moderate salinization soil (0.4–0.6%), and severe salinization soil (0.6–1.0%) (*Wang et al., 2018a*). The statistical analysis results of SSC are shown in Table 2. The percentages of mild salinization soil, moderate salinization soil, and severe salinization soils were 25.0%, 65.0%, and 10.0%, respectively (Table 2), and their coefficient of variation was 0.20, indicating that the salt content of the soil surface had moderately weak variability.

## Sensitive variable selection

The correlation between spectral index and SSC was analyzed by a correlation analysis system in SPSS21.0, and the obtained correlation coefficients are shown in Table 3. The spectral index were significantly correlated with SSC at the $p < 0.001$ level (SI4 was significantly correlated with SSC at the $p < 0.05$ level), except for the spectral indexes EVI and COSRI, which did not correlate with SSC (Table 3). To screen the sensitive variables, we selected the spectral index which significantly correlated with SSC ($p < 0.001$) and set the correlation coefficient selection threshold to | 0.8 |. The sensitive spectral indexes screened by PCC were NDVI, PNDVI, SAVI, MSAVI, ARVI, OSAVI, NDSI, S3, S4, S5, SI1, SI3, and SI.

The relationship between spectral index and SSC was also analyzed by GRA and VIP in MATLAB R2016a (Fig. 3). The GCD selection threshold was set to 0.7, and the sensitive spectral indexes screened by GRA were DVI, EVI, TVI, GNDVI, MSAVI, CRSI, RDVI, BI, S1, S3, S4, S6, SI1, SI2, SI3, SI4, SI, SI-T, and COSRI. Conversely, the VIP selection threshold was set to 1.0, and the sensitive spectral indexes screened by VIP were NDVI, GNDVI, PNDVI, SAVI, MSAVI, ARVI, OSAVI, NDSI, BI, S3, S4, S5, S6, SI1, SI2, SI3, and SI. The sensitive spectral indexes screened by SVM-RFE were NDVI, NDGI, GNDVI, PNDVI, SAVI, MSAVI, ARVI, OSAVI, NDSI, S3, S4, S6, SI1, and SI.

In summary, the PCC, GRA, VIP, and SVM-RFE variable selection methods screened 13, 19, 17, and 14 sensitive spectral indexes, respectively, indicating that variable selection can effectively remove redundant information of spectral parameters and improve the accuracy of soil salinity inversion models.

**Table 3 The correlation coefficient between spectral index and SSC.**

| Spectral index | Correlation coefficient | Spectral index | Correlation coefficient | Spectral index | Correlation coefficient |
|---|---|---|---|---|---|
| NDVI | −0.82** | MSAVI | −0.82** | S4 | 0.87** |
| DVI | −0.39** | CRSI | −0.68** | S5 | 0.86** |
| EVI | −0.23 | ARVI | −0.80** | S6 | 0.79** |
| RVI | −0.65** | RDVI | −0.69** | SI1 | 0.87** |
| TVI | −0.41** | OSAVI | −0.82** | SI2 | 0.74** |
| NDGI | −0.70** | NDSI | 0.82** | SI3 | 0.86** |
| GNDVI | −0.78** | BI | 0.67** | SI4 | −0.31* |
| PNDVI | −0.80** | S1 | −0.52** | SI | 0.87** |
| GVI | −0.65** | S2 | −0.52** | SI-T | 0.39** |
| SAVI | −0.82** | S3 | 0.86** | COSRI | −0.2 |

**Note:**
*For significance test $p < 0.05$, **for $p < 0.01$.

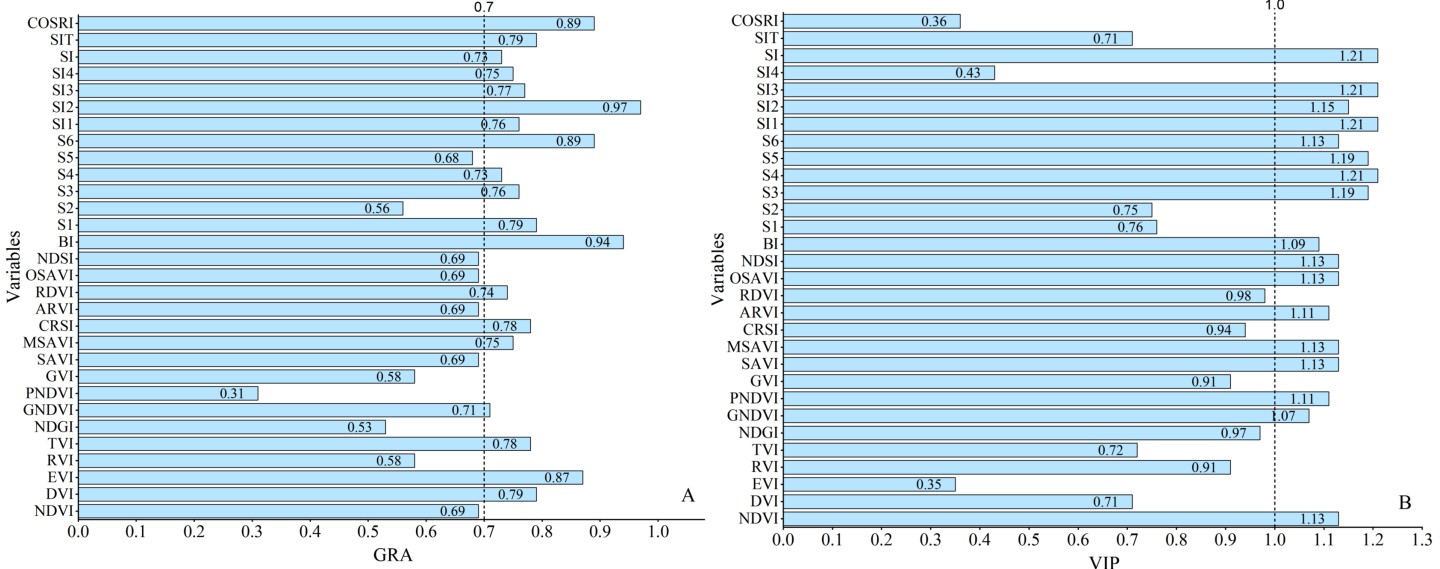

**Figure 3 The analysis results of spectral index and SSC under different sensitive variable selection methods.** (A) GRA analysis results. (B) VIP analysis results.

## Construction of soil salinity inversion model

### Soil salinity inversion model based on the PCC

The results of soil salinity inversion model based on PCC are shown in Table 4. Based on the comprehensive analysis of the modeling set $R_c^2$ and $RMSE_c$ and the validation set $R_v^2$, $RMSE_v$, and RPD, the SVM model demonstrated to be the best salinity inversion model under the PCC variable selection method, with the $R_c^2$, $R_v^2$, $RMSE_v$, and RPD values of 0.8854, 0.8539, 0.1634, and 1.9738, respectively. The other models had smaller differences in their accuracy, with their $R_c^2$ being between 0.7807 and 0.8098, $R_v^2$ between 0.7615 and 0.8513, RMSE below 0.2, and RPD above 1.8, thus indicating a good inversion effect.

**Table 4 The soil salinity inversion model based on PCC.**

| Variable selection method | Algorithm | Modeling set | | Verification set | | RPD |
|---|---|---|---|---|---|---|
| | | $R_c^2$ | RMSEc | $R_v^2$ | RMSEv | |
| PCC | SVM | 0.8854 | 0.1050 | 0.8539 | 0.1634 | 1.9738 |
| | BPNN | 0.8098 | 0.1816 | 0.7615 | 0.1385 | 2.3280 |
| | ELM | 0.8096 | 0.1463 | 0.8005 | 0.1703 | 1.8935 |
| | RF | 0.7807 | 0.1591 | 0.8513 | 0.1428 | 2.2613 |

**Table 5 The soil salinity inversion model based on GRA.**

| Variable selection method | Algorithm | Modeling set | | Verification set | | RPD |
|---|---|---|---|---|---|---|
| | | $R_c^2$ | RMSEc | $R_v^2$ | RMSEv | |
| GRA | SVM | 0.7863 | 0.1652 | 0.8888 | 0.1780 | 1.8115 |
| | BPNN | 0.7997 | 0.1844 | 0.8040 | 0.2946 | 1.7945 |
| | ELM | 0.7967 | 0.1511 | 0.8634 | 0.1852 | 1.7411 |
| | RF | 0.8004 | 0.1681 | 0.7915 | 0.1770 | 1.8214 |

**Table 6 The soil salinity inversion model based on VIP.**

| Variable selection method | Algorithm | Modeling set | | Verification set | | RPD |
|---|---|---|---|---|---|---|
| | | $R_c^2$ | RMSEc | $R_v^2$ | RMSEv | |
| VIP | SVM | 0.8986 | 0.1114 | 0.8447 | 0.1699 | 1.8979 |
| | BPNN | 0.8617 | 0.1271 | 0.7530 | 0.1620 | 1.9899 |
| | ELM | 0.8396 | 0.1343 | 0.8014 | 0.1732 | 1.8615 |
| | RF | 0.7978 | 0.1580 | 0.8264 | 0.1511 | 2.1346 |

### Soil salinity inversion model based on the GRA

The results of soil salinity inversion model based on GRA are shown in Table 5. The comprehensive analysis of the modeling set ($R_c^2$ and $RMSE_c$) and validation set ($R_v^2$, $RMSE_v$ and RPD) showed that the SVM was the best salinity inversion model under the GRA variable selection method, with $R_c^2$, $R_v^2$, $RMSE_v$, and RPD values of were 0.7864, 0.8888, 0.1780, and 1.8115, respectively. The other models exhibited slight differences in their accuracy, with the $R_c^2$ ranging between 0.7967 and 0.8004, $R_v^2$ between 0.7915 and 0.8634, RMSE below 0.3, and RPD above 1.7, indicating a good inversion effect.

### Soil salinity inversion model based on the VIP

The results of soil salinity inversion model based on VIP are shown in Table 6. Like PCC and GRA selection methods, the SVM was the best salinity inversion model under the VIP variable selection method, with $R_c^2$, $R_v^2$, $RMSE_v$, and RPD values of 0.8986, 0.84470.1699, and 1.8979, respectively. There were slight accuracy differences in the other models, with

**Table 7 The soil salinity inversion model based on SVM-RFE.**

| Variable selection method | Algorithm | Modeling set | | Verification set | | RPD |
|---|---|---|---|---|---|---|
| | | $R_c^2$ | RMSEc | $R_v^2$ | RMSEv | |
| SVM-RFE | SVM | 0.8994 | 0.1835 | 0.8674 | 0.2170 | 1.9863 |
| | BPNN | 0.8686 | 0.1259 | 0.7415 | 0.1797 | 1.7938 |
| | ELM | 0.8855 | 0.1134 | 0.8084 | 0.1820 | 1.7714 |
| | RF | 0.7879 | 0.1579 | 0.8144 | 0.1502 | 2.1464 |

$R_c^2$ ranging between 0.7978 and 0.8617, $R_v^2$ between 0.7530 and 0.8264, RMSE below 0.2, and RPD above 1.8, thus showing a good inversion effect.

### Soil salinity inversion model based on the SVM-RFE

The results of soil salinity inversion model based on SVM-RFE are shown in Table 7. The $R_c^2$, $R_v^2$, $RMSE_v$, and RPD values of 0.8994, 0.8674, 0.2170, and 1.9863, respectively, showed that the SVM model was the best salinity inversion model under the SVM-RFE variable selection method. The other models had smaller differences in their accuracy, with their $R_c^2$ ranging between 0.7879 and 0.8855, $R_v^2$ between 0.7415 and 0.8144, RMSE below 0.2, and RPD above 1.7, suggesting a good inversion effect.

## Comprehensive evaluation analysis of soil salinity inversion models

The 20 soil salinity inversion models were constructed with the spectral index as the model input independent variable and the corresponding soil salinity as the dependent variable. The results of the modeling set and validation set analysis are shown in Table 8 and Fig. 4.

The $R_c^2$ and $R_v^2$ values of the inversion models generated using the same machine learning algorithm based on PCC, GRA, VIP, and SVM-RFE variable selection methods were larger than that of CK (without variable selecting) but were closer to each other (Table 8), indicating that the models did not exhibit "overfitting" phenomenon. Conversely, the $RMSE_c$ and $RMSE_v$ of the models were smaller than CK (except for $RMSE_v$ of the SVM-RFE-SVM model), and their RPD was above 1.7, indicating that the soil salinity inversion model had a good inversion effect and can improve the accuracy of the inversion models. For the SVM, BPNN, and ELM algorithms, the inversion model $R_v^2$ based on the GRA variable selection method was larger than that of PCC, VIP, and SVM-RFE, while for the RF algorithm, the inversion model $R_v^2$ based on the PCC variable selection method was larger than that of GRA, VIP, and SVM-RFE. This shows that the GRA variable selection method is suitable for SVM, BPNN, and ELM modeling, while the PCC method is suitable for RF modeling.

Since the differences in RMSE values were small among the models, the evaluation index of the validation set ($R_v^2$, $RMSE_v$, and RPD) were used as parameters to draw the evaluation index of the accumulation bar chart for each model (Fig. 5) to distinguish the models accurately, and further evaluate the inversion effect of each model.

**Table 8 The soil salinity inversion models based on different variable selection methods and machine learning algorithms.**

| Variable selection method | Algorithm | Modeling set | | Verification set | | RPD |
|---|---|---|---|---|---|---|
| | | $R_c^2$ | $RMSE_c$ | $R_v^2$ | $RMSE_v$ | |
| CK | SVM | 0.7801 | 0.2023 | 0.8319 | 0.2000 | 1.7697 |
| PCC | | 0.8854 | 0.1050 | 0.8539 | 0.1634 | 1.9738 |
| GRA | | 0.7863 | 0.1652 | 0.8888 | 0.1780 | 1.8115 |
| VIP | | 0.8986 | 0.1114 | 0.8447 | 0.1699 | 1.8979 |
| SVM-RFE | | 0.8994 | 0.1835 | 0.8674 | 0.2170 | 1.9863 |
| CK | BPNN | 0.6819 | 0.2057 | 0.6870 | 0.3428 | 1.4407 |
| PCC | | 0.8098 | 0.1816 | 0.7615 | 0.1385 | 2.3280 |
| GRA | | 0.7997 | 0.1844 | 0.8040 | 0.2946 | 1.7945 |
| VIP | | 0.8617 | 0.1271 | 0.7530 | 0.1620 | 1.9899 |
| SVM-RFE | | 0.8686 | 0.1259 | 0.7415 | 0.1797 | 1.7938 |
| CK | ELM | 0.7852 | 0.1517 | 0.7391 | 0.1915 | 1.6840 |
| PCC | | 0.8096 | 0.1463 | 0.8005 | 0.1703 | 1.8935 |
| GRA | | 0.7967 | 0.1511 | 0.8634 | 0.1852 | 1.7411 |
| VIP | | 0.8396 | 0.1343 | 0.8014 | 0.1732 | 1.8615 |
| SVM-RFE | | 0.8855 | 0.1134 | 0.8084 | 0.1820 | 1.7714 |
| CK | RF | 0.7702 | 0.1687 | 0.7785 | 0.1860 | 1.7937 |
| PCC | | 0.7807 | 0.1591 | 0.8513 | 0.1428 | 2.2613 |
| GRA | | 0.8004 | 0.1681 | 0.7915 | 0.1770 | 1.8214 |
| VIP | | 0.7978 | 0.1580 | 0.8264 | 0.1511 | 2.1346 |
| SVM-RFE | | 0.7879 | 0.1579 | 0.8144 | 0.1502 | 2.1464 |

As shown in Fig. 5A, the length of the model bar chart, based on the variable selection methods, was greater than that of CK, indicating that the accuracy of the soil salinity inversion models can be improved by variable selection. The length difference of the model bar chart generated based on the GRA variable selection method was small, while that based on the PCC, VIP, and SVM-RFE variable selection methods was large under the same machine learning algorithm. This showed that the GRA is the best sensitive variable selection method. Moreover, as shown in Fig. 5B, the length difference of the model bar chart based on SVM was small, while that based on BPNN, ELM, and RF was large under the same variable selection method, indicating that the SVM modelwas the best soil salinity inversion model. Through comprehensive analysis of the bar graph length of each model, we found that the differences in inversion model results were due to the different variable selection methods, indicating the importance of selecting appropriate variable selection methods for improving the accuracy of the model.

The SSC in the study area based on the GRA-SVM model inversion is shown in Fig. 6. As shown in Fig. 6, the spatial distribution map of soil salinity based on GRA-SVM inversion model can truly reflect the degree of soil salinization in the study area. The study can provide a theoretical basis for the effective prevention and control of soil salinization in this area.

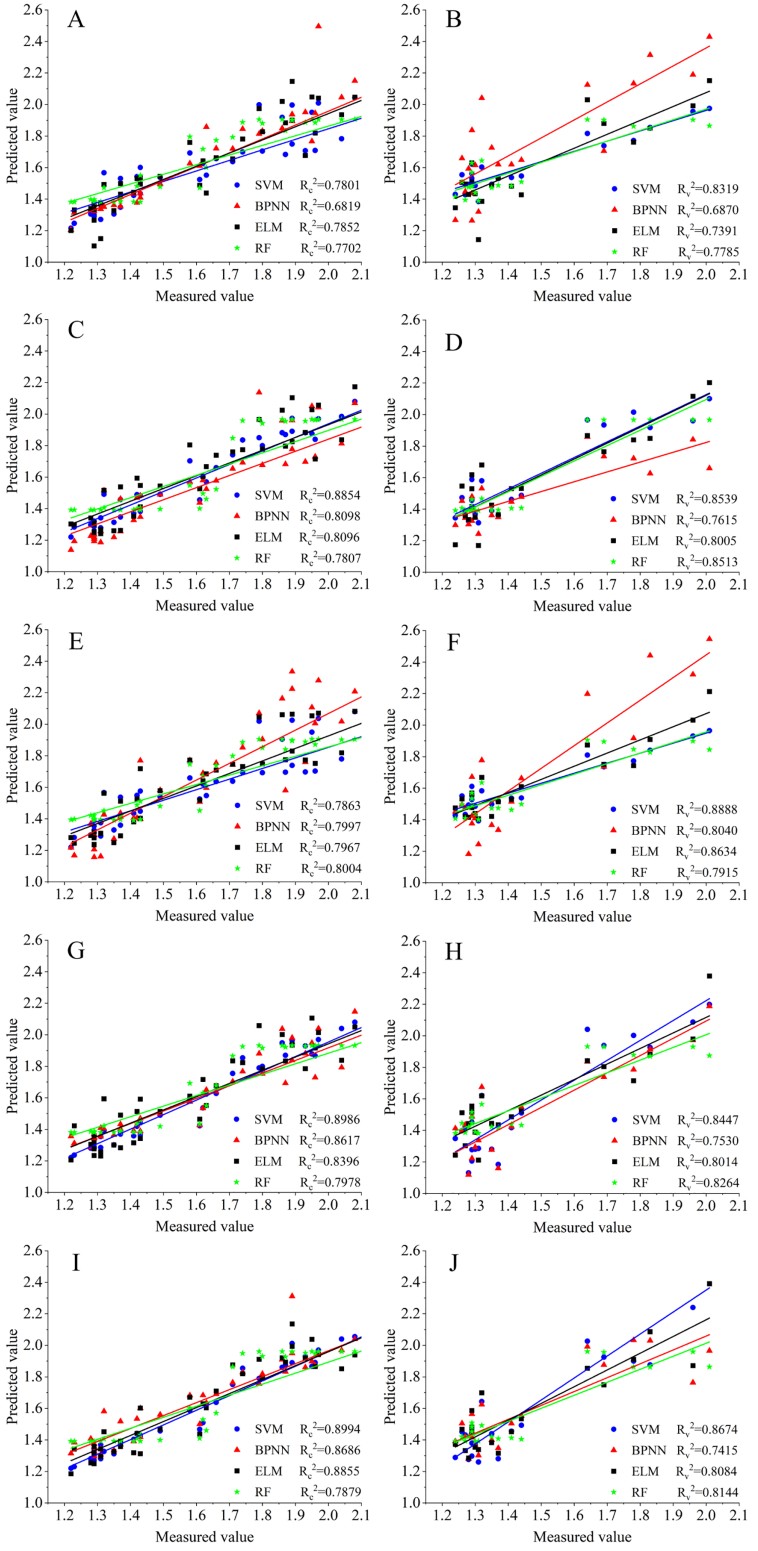

**Figure 4 Comparison of measured and predicted soil salinity.** (A) CK modeling. (B) CK verification. (C) PCC modeling. (D) PCC verification. (E) GRA modeling. (F) GRA verification. (G) VIP modeling. (H) VIP verification. (I) SVM-RFE modeling. (J) SVM-RFE verification.

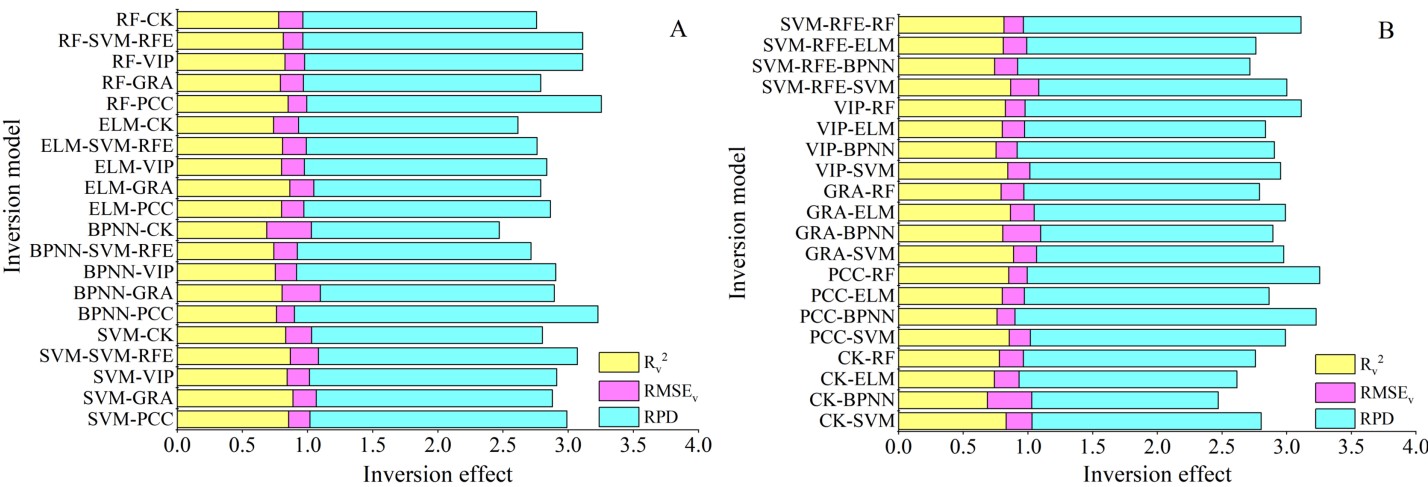

**Figure 5 The stacked bar graphs of indicators for comprehensive evaluation of different soil salinity inversion models.** (A) The different selection methods under the same algorithm. (B) The different algorithms under the same selection method.

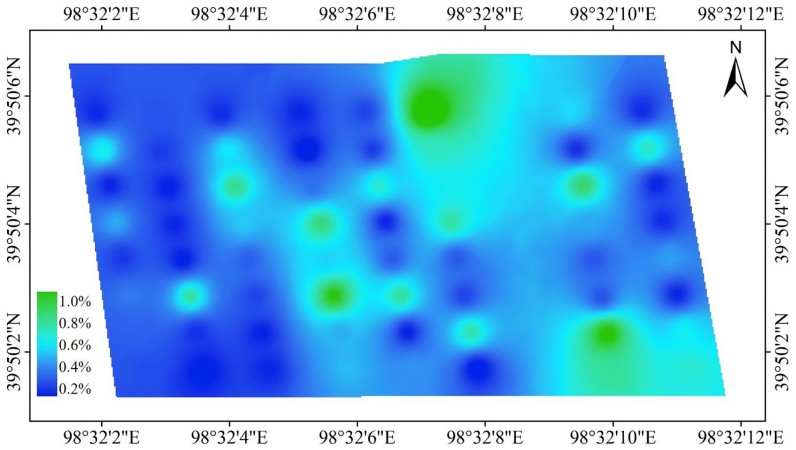

**Figure 6 SSC spatial distribution map based on GRA-SVM inversion model.**

## DISCUSSION

UAV remote sensing has great development potential in estimating soil salinity. Based on its fast information acquisition capability, wide coverage, and low operating cost, UAV remote sensing also represents the future development direction of precision agriculture (*Khanal, Fulton & Shearer, 2017*; *Rejeb et al., 2022*). This study utilizes UAV remote sensing technology to obtain rich spectral information. Using this information, it constructs a alfalfa covered soil salinity inversion model based on variable selection and machine learning algorithms, significantly improving the accuracy of soil salinity inversion. By selecting sensitive variables, redundant information of spectral parameters can be effectively removed, improving the accuracy of soil salinity inversion. The combination of variable selection method and machine learning algorithm further

improves the accuracy of soil salinity inversion. This study provide a new approach for timely and accurate acquisition of crops covered farmland soil salt information.

The use of numerous spectral variables complicates the construction of the selection model due to information redundancy associated with such an amount of spectral variables. Therefore, selecting appropriate variable selection methods is an effective way of improving the monitoring accuracy of soil salt content (*Maes & Steppe, 2019*; *Ivushkin et al., 2019*; *Wang et al., 2018b*; *Jang et al., 2022*). This study constructed 30 spectral indexes using UAV multispectral remote sensing data and screened 13, 19, 17, and 14 sensitive spectral indexes based on PCC, GRA, VIP, and SVM-RFE, respectively. Thus, GRA screened the highest, while PCC screened the lowest number of sensitive spectral index. These results show that the variable selection method can effectively remove redundant information in the spectral variables, reduce the complexity of the salinity inversion model, and improve the accuracy of the soil salinity inversion model. However, each variable selection method exhibits different model optimization, as reported by previous studies (*Cao et al., 2020*; *Miao, Li & Lu, 2018*; *Han et al., 2019*; *Mwinuka et al., 2022*; *Jia et al., 2020*). A comprehensive comparative analysis of the accuracy of the inversion model based on the variable selection methods showed that GRA was the best among the four sensitive variable selection methods, consistent with the findings of previous studies (*Zhao et al., 2023*; *Jia et al., 2020*). This is mainly due to the small amount of calculation of GRA, applicable with multiple samples, and exhibits no discrepancies between quantitative and qualitative analysis results; thus can better reflect the correlation between variables. *Wang et al. (2019a)* used GRA, VIP, and stepwise regression analysis (SR) to screen the sensitive variables and showed that the model accuracy based on VIP variable selection method was the highest. Furthermore, *Wang et al. (2019b)* used PCC, VIP, GRA, and random forest (RF) to the screen variables and found that the model inversion effect of the RF variable selection method was the best. This may be due to the differences in soil texture, meteorological environment, and planting structure among the study subjects.

In addition, due to the influence of soil environment, anthropogenic causes and other factors, the independent variables and dependent variables are in a nonlinear relationship, and the predictive ability of the crops covered farmland soil salinity inversion model constructed based on the linear regression algorithm will not be able to meet the accuracy requirements (*Leroux et al., 2019*; *Periasamy, Ravi & Tansey, 2022*). Therefore, this study constructed 20 crops covered farmland soil salinity inversion models by using SVM, BPNN, ELM and RF four machine learning algorithms based on the spectral index screened by the sensitive variable selection method. The results showed that the overall inversion effect of the salt inversion model based on the four sensitive variable selection methods was better than that of the unscreened, and the RPD of the model is above 1.7, indicating that the process of selection the spectral index by the sensitive variable selection method can further improve its correlation with salt, thereby improving the accuracy of the salt inversion model. The results also showed that under the same sensitive variable selection method, different modeling methods are selected, and the effect of salt inversion model is different. It is pointed out that the salt inversion model based on support vector machine (SVM) is the best inversion model, which is consistent with the research results of

*Jia et al. (2020)*, *Feng et al. (2018)*. By comparing the inversion accuracy of different machine learning algorithms, *Zhang et al. (2019b)*, *Ge et al. (2019)* and *Wei et al. (2020)* found that random forest (RF) achieved the highest inversion accuracy. *Liu et al. (2022)* pointed out that BP neural network (BPNN) is the best soil salinity inversion model. This is mainly due to the complex composition of soil salinity and strong spatial variability, so that different algorithms may have different inversion effects in different regions.

## CONCLUSIONS

In this study, we evaluated the feasibility of different variable selection methods combined with machine learning algorithms in improving the accuracy of SSC inversion in crop-covered farmland, and reached the following conclusions:

1. The RPD values of all models were greater than 1.7, indicating all models have an ability to quantitatively estimate SSC, so the variable selection combined with machine learning algorithm is an effective method to improve the accuracy of soil salinity remote sensing inversion.

2. The performance of the models has been improved markedly using the four variable selection methods, and the applicability varied among the four methods: the GRA variable selection method is suitable for SVM, BPNN, and ELM modeling, while the PCC method is suitable for RF modeling.

3. The choice of different variable selection combined with machine learning algorithm had a great effect on the prediction accuracy of the model. The GRA-SVM is the best soil salinity inversion model in alfalfa cover farmland, with $R_v^2$ of 0.8888, $RMSE_v$ of 0.1780, and RPD of 1.8115 based on the model verification dataset, and the spatial distribution map of soil salinity can truly reflect the degree of soil salinization in the study area.

By comparative research, this study found that different variable selection methods can effectively improve the prediction accuracy of soil salinity, and are not affected by sampling time, location and differences in spectral reflectance. At the same time, this study further compared and verified the applicability of variable selection methods combined with machine learning algorithms in soil salinity inversion, and determined the characteristic variables suitable for soil salinity inversion in alfalfa covered farmland in arid oasis areas, as well as the optimal variable selection method and soil salinity inversion model. Although the models in the study performed satisfactorily in the inversion of SSC, their accuracy needs further improving. In future research, the feasibility of coupling different variable method methods to improve model prediction accuracy will be further explored, as well as the differences in other variable method methods in improving model performance, increasing computational efficiency, and reducing the impact of redundant information.

## ACKNOWLEDGEMENTS

We thank the Taolai River Basin Water Resources Utilization Center, Gansu Provincial Department of Water Resources for providing the vector files of the study area. We are especially grateful to the editors and reviewers for appraising our manuscript and for offering instructive comments.

### Funding

This work was supported by The National Natural Science Foundation of China (52379042), the Jiuquan Science and Technology Program (2023CA2060), and the Key Projects of Jiuquan Vocational and Technical College (2023XJZXM01). The funders had no role in study design, data collection and analysis, decision to publish, or preparation of the manuscript.

### Grant Disclosures

The following grant information was disclosed by the authors:
National Natural Science Foundation of China: 52379042.
Jiuquan Science and Technology Program: 2023CA2060.
Jiuquan Vocational and Technical College: 2023XJZXM01.

### Competing Interests

The authors declare that they have no competing interests.

### Author Contributions

- Hong Ma conceived and designed the experiments, performed the experiments, analyzed the data, prepared figures and/or tables, authored or reviewed drafts of the article, and approved the final draft.
- Wenju Zhao conceived and designed the experiments, performed the experiments, analyzed the data, prepared figures and/or tables, authored or reviewed drafts of the article, and approved the final draft.
- Weicheng Duan conceived and designed the experiments, performed the experiments, analyzed the data, prepared figures and/or tables, authored or reviewed drafts of the article, and approved the final draft.
- Fangfang Ma conceived and designed the experiments, performed the experiments, analyzed the data, prepared figures and/or tables, authored or reviewed drafts of the article, and approved the final draft.
- Congcong Li conceived and designed the experiments, performed the experiments, analyzed the data, prepared figures and/or tables, authored or reviewed drafts of the article, and approved the final draft.
- Zongli Li conceived and designed the experiments, analyzed the data, prepared figures and/or tables, authored or reviewed drafts of the article, and approved the final draft.

### Data Availability

The raw measurements are available in the Supplemental Files.

### Supplemental Information

Supplemental information for this article can be found online at http://dx.doi.org/10.7717/peerj.18186#supplemental-information.

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
