# Peer review of "Inversion model of soil salinity in alfalfa covered farmland based on sensitive variable selection and machine learning algorithms"

_PeerJ, doi:10.7717/peerj.18186_

## Round 0.1 · original submission · Major Revisions

Both reviewers found the manuscript has the quality for publication. However, major concerns regarding the originality of the manuscript need to be addressed. As pointed out by the first reviewer, the authors' previous works have high correlation with the current one. Therefore, it is mandatory to clarify the main innovations of this work over the previous ones.
Furthermore, the manuscript tends to make many self-citations, which might reduce the credibility of this work. For example, the methods used in the work including PCC, GRA, VIP, SVM-RFW should be referenced to the original publications rather than the works who used them.
In addition to the comments of the reviewers, the clarity and conciseness of the content shall be improved.

Reviewer 1 ·

Basic reporting

This research paper focuses on improving the accuracy of soil salinity inversion models by combining sensitive variable selection methods and machine learning algorithms. Methods such as Pearson correlation coefficient (PCC), gray ratio analysis (GRA), variance importance (VIP), and support vector machine-based recursive feature elimination (SVM-RFE) were used to select the most relevant variables. Salinity inversion models were built using algorithms such as support vector machine (SVM) regression, backpropagation neural network (BPNN), extreme learning machine (ELM), and random forest (RF). The results demonstrated that the combination of variable selection methods with machine learning algorithms significantly improves the accuracy of soil salinity monitoring, with the GRA-SVM model being the most effective.

Although the work is well developed and its motivations and objectives are clear, below I advise some changes, additions or corrections that could help improve the impact of the research paper.

1. The title of the paper includes the word UAV, this suggests that the use of UAV is an important variable and perhaps one of the most important in the development of this research work. However, the UAV is only the means to transport the multispectral sensor. I advise including more information about the impact and variability included in the use of this sensor transport system to justify the use of this word in the title and not end up creating false expectations.

2. The methods section in the abstract could be rewritten, better explaining the methodological sequence of how the proposed method was developed. It is difficult to understand the proposed methodology since in the abstract it looks like a list of methods. (This could be improved)

3. After line 82 a paragraph should be added with a comparative analysis of the model proposed in this research work and the works cited in the state of the art. This paragraph should emphasize the differences and novelties offered by the proposed method and highlight the importance of what this paper proposes.

4. The paragraph that begins on line 83 should focus the description of the proposed method on the contributions that the proposed method makes to the state of the art of the research area. (Not only limit the description of the research in terms of the objectives, but also emphasize the contributions and factors that make this research work unique)

Experimental design

5. The UAV was used to transport the sensor, but there is no discussion of the effects that this may entail, the height of the UAV and the consequences on the measurements, the speed of the UAV and the effects on data collection, the area of sample coverage vs the height at which the sample is taken, the UAV pathplaning algorithm that ensures total coverage of the area in relation to the speed of the UAV and its height. Percentage of overlaping in the measurements.
There are many parameters inherent to the effect of the UAV in this study. Since the paper only focuses on the sensor and the measurements carried out, completely ignoring the effect of the UAV, I would advise removing the word UAV from the title and simply mentioning in the abstract and in "research methods" that a UAV was used for the recording. of samples in this study.

6. It is necessary to indicate what work was carried out for this new paper and what was previously carried out in the 4 works of the main author and which are highly correlated with the present research work (The novelty of this paper is seriously compromised by taking into account previous works):

Zhao W, Zhou C, Zhou C, Ma H, Wang Z. 2022. Soil salinity inversion model of oasis in arid area based on UAV multispectral remote sensing. Remote Sens 14(8): 1804. DOI 10.3390/rs14081804

Zhao W, Ma H, Zhou C, Zhou C, Li Z. 2023. Soil salinity inversion model based on BPNN optimization algorithm for UAV multispectral remote sensing. IEEE J Sel Top Appl Earth Obs Remote Sens (16): 6038-6047. DOI 10.1109/jstars.2023.3284019

Zhao W, Ma F, Ma H, Zhou C. 2022. Soil salinity inversion model based on the multispectral images of UAV. Transactions of the CSAE 38(24):93-101. DOI 10.11975/j.issn.1002-6819.2022.24.010

Zhao W, Duan W, Wang Y, Zhou C, Ma H. 2023. Multispectral vegetation water content inversion model based on sensitive variable screening. Trans Chin Soc Agric Mach 54(9):343-351+385.

Validity of the findings

7. It is necessary to contrast the new findings with those obtained in the previous works of the pre-principal Author (the 4 mentioned previously).

8. The conclusions should include a short paragraph indicating the direction that could guide future work in this area or the next step for this same working group in this area of ​​research.

Additional comments

9. The color map bar in figure 6 should be along the y axis to show the values ​​related to the color range in better detail. Include the bar in the graph on the right side along the y axis (Vertical), this would help to better appreciate the meaning of the color variation and the real values ​​that this means.

Reviewer 2 ·

Basic reporting

• The problem is not well stated in the introduction.
- The required format of the journal is not fully respected. Eg : the titles of the article and the authors' affiliations, the author used Calibri format instead of Arial, and the same applies to the body of the article. The same issue exists with the subtitles etc. The format needs to be revised.
• The tables and figures are well presented and clear, but I suggest adding a clear photo of the study site.
• It is necessary to correct and verify the way authors are cited in the text, including parentheses, etc.
• Use simpler sentences.
• Add more details on the spatial distribution of the 64 sampling points.

Experimental design

No comment

Validity of the findings

No comment

Additional comments

I recommend changing this title. Please do not start the title with with 'UAV'.Try to make it easy to understand and to find when searching online.

I consider this article to be of good quality and to contain significant results. It would be beneficial to publish it after making some improvements.

---

## Round 0.2 · Minor Revisions

Please follow the reviewer's final comments to improve the manuscript.

Reviewer 1 ·

Basic reporting

Overall: The paper is now more organized and focused on the content that is really intended to be shown and the results that are intended to be emphasized.
The new references included help to locate the paper within the state of the art.

Experimental design

The added paragraphs allow us to better understand the approach that the authors followed within the experimental process.
1. Despite the additions and explanations made to the experimental part, a rather strong overlap with the authors' previous works can still be appreciated.

Validity of the findings

2. Much more emphasis needs to be placed on the findings of both the project in general and those findings that differentiate this paper from the authors' previous publications. The section contrasting current results and the improvement that this article represents needs to be further strengthened.

Additional comments

Overall, the paper is much improved from the original version.
It is now possible to better appreciate the objectives (contributions) of the present work as well as to find out why this apparent conclusive work of the project was necessary.
Note: Thank you for attending to most of the suggested changes and additions.

---

## Round 0.3 · accepted · Accept

You have addressed all the reviewer's comments. The conclusion has been rewritten with strong statements. I am happy with the current version. Please go through the manuscript to remove some minor grammatic problems, such as "indexs" line 83.